# A Review of the Density, Biomass, and Secondary Production of Odonates

**DOI:** 10.3390/insects15070510

**Published:** 2024-07-08

**Authors:** Anais Rivas-Torres, Adolfo Cordero-Rivera

**Affiliations:** Universidade de Vigo, ECOEVO LAB, Escola de Enxeñaría Forestal, Campus Universitario A Xunqueira s/n, 36005 Pontevedra, Spain; adolfo.cordero@uvigo.gal

**Keywords:** ecosystem services, larval odonata, matter exportation

## Abstract

**Simple Summary:**

Dragonflies and damselflies are invaluable components of freshwater ecosystems, acting as dominant predators and facilitating the exportation of matter and energy from aquatic to terrestrial environments thanks to their powerful flight. They are also crucial as a food source for various animals and, in some cases, for humans. Through a comprehensive review of the literature, we estimated the biomass, density, and secondary production of these insects, assessing their potential significance in terrestrial fertilization. Our findings indicate that dragonfly larvae are particularly abundant in lentic habitats. Overall, the evidence suggests that dragonflies and damselflies may make a substantial contribution to the exportation of materials to terrestrial systems, especially considering the adults’ ability to migrate and inhabit different types of water ecosystems.

**Abstract:**

Freshwater insects are highly significant as ecosystem service providers, contributing to provisioning services, supporting services, and cultural services. Odonates are dominant predators in many freshwater systems, becoming top predators in fishless ecosystems. One service that odonates provide is the export of matter and energy from aquatic to terrestrial ecosystems. In this study, we provide a review of the literature aiming to estimate the density, biomass, and secondary production of odonates and discuss to what extent this order of insects is relevant for the fertilization of terrestrial ecosystems. We found published data on 109 species belonging to 17 families of odonates from 44 papers. Odonata larvae are abundant in freshwater systems, with a mean density of 240.04 ± 48.01 individuals m^−2^ (±SE). Lentic habitats show much higher densities (104.40 ± 55.31 individuals m^−2^, N = 118) than lotic systems (27.12 ± 5.09, N = 70). The biomass estimations for odonates indicate values of 488.56 ± 134.51 mg m^−2^ y^−1^, with similar values in lentic and lotic habitats, which correspond to annual secondary productions of 3558.02 ± 2146.80 mg m^−2^ y^−1^. The highest biomass is found in dragonflies of the Aeshnidae, Corduliidae, and Gomphidae families. The available evidence suggests a significant potential contribution of Odonata to the exportation of material from water bodies to land. This is further strengthened by the ability of adult odonates to migrate and to colonize different types of water bodies.

## 1. Introduction

The properties of ecosystems at the landscape level are conditioned by the biotic interactions among organisms but also by abiotic factors and constraints. This point of view has emerged as the meta-ecosystem concept, which has superseded the previous metacommunity concept [1], which focused on biotic components. The meta-ecosystem is a set of ecosystems connected by flows of energy and materials, many of which are mediated by the dispersal of organisms, although some are not [2].

Aquatic systems, particularly lotic ecosystems, receive large amounts of matter and energy from the terrestrial ecosystems of their drainage basin, therefore becoming good indicators of the conservation status of the terrestrial systems [3]. In fact, the relationship between rivers and their basins is a central topic in river ecology [4]. However, how allochthonous inputs (i.e., subsidies) from aquatic systems affect land ecosystems is understudied, although there is clear evidence that food webs and the productivity of the terrestrial ecosystems are affected by freshwater aquatic subsidies [5,6,7,8] or even marine subsidies [9,10]

The ecological linkage between riparian ecosystems and freshwaters has been reviewed considering two pathways [11,12,13]: (I) abiotic coupling, that is, via flood or drought events; and (II) biotic coupling, which means via the emergence of merolimnic aquatic insects. Freshwater insects are highly significant as ecosystem service providers, contributing to provisioning services, supporting services, and cultural services [14]. Some studies have shown that the emergence of aquatic insects from streams can make up a considerable export of the benthic production to riparian predators like spiders, lizards, birds, frogs, and bats, conferring up to 25–100% of the energy or carbon to such species [12]. A few field experiments that manipulated this subsidy showed that it affected the short-term behavior, growth, and abundance of terrestrial consumers. For instance, Nakano and Murakami [15] described that aquatic insects contribute to 26% of the total annual energy budget of the entire bird assemblage, becoming the predominant food in autumn and providing 98% of bird food during winter. Furthermore, Iwata et al. [16] and Henschel et al. [17] found that aquatic prey constituted 67–82% of bird diets and 54% of spider diets along the lotic systems.

In this context, behavior during the emergence of aquatic insects, such as concentrations in time and space, can alter the behavior and demographics of migratory consumers, resulting in an increased abundance during peak emergence periods. To summarize, the general pattern from these studies [15,18,19,20] is that emergent aquatic insects attract consumers to riparian zones along streams, principally during the peak emergence in spring or early summer, either for their entire life via colonization as occurs in spiders and beetles or temporarily via immigration as for birds or bats.

Aquatic insects export a significant amount of biomass to terrestrial systems [5,21]. In the case of agriculture, winged insects contribute to pollination and soil fertilization, and odonates to pest control due to their carnivorous behavior [22]. Odonates are dominant predators in many freshwater systems, becoming the top predators in fishless ecosystems. In fact, a review of the density of benthic macroinvertebrates in streams in North and South America showed that odonates were the dominant predatory taxon on 96% of the sites examined [23]. Odonates are considered to be of little economic importance, because they are not pests and do not affect human health in a significant way, although they might sometimes suppress mosquitoes or other insects of medical importance [24,25] and they do provide aesthetic and spiritual benefits to humans [26]. 

The importance of odonates as bioindicators of water quality [27] or as model species for ecology and evolution is widely recognized, in part due to their widespread distribution and conspicuousness [28], as well as the fact that they can be studied by exuviae collection, making the assessments environmentally friendly because no individuals have to be collected [29]. In this review, we concentrate on the role of odonates as ecosystem services providers in terms of matter and energy export (density, biomass, and secondary production) from aquatic to terrestrial ecosystems, a topic that has received little attention. 

## 2. Materials and Methods

The published data on density, biomass, and secondary production were searched by querying Google Scholar for “secondary production and odonat*”, “biomass and odonat*”, “emergence, larva and odonat*”, and “density and odonat*”. The searches were carried out until May 2024. The selection process of the papers to be included in this review was divided into four steps: (1) identification, where we collated all the available information about the density, biomass, and secondary production of Odonata in different environments; (2) first review, where we then read the abstracts to select the papers that included the keywords mentioned above to check if they potentially provided the information needed; (3) second review, where we read in their entirety all the articles chosen in the previous step and extracted the data and information necessary for our study; and (4) obtaining some of the data through calculations of information included in the papers and others through estimation of the numerical data from the figures (see below for details).

We compiled information for a total of 109 species, genera, or families, or for the Odonata as a whole, from 44 papers. The full details of the literature reviewed, the variables considered, and the species included in the analyses are given in the dataset presented in Appendix A. 

For the purposes of our study, we compiled the following set of variables from each study: taxonomic unit (species if indicated), family, suborder, habitat (lentic or lotic), density, biomass, secondary production, and the locality. The latitude (as an absolute value), longitude, and altitude were obtained from the papers or estimated by searching the location in Google Earth. The response variables were standardized: density (number m^−2^), biomass (mg m^−2^ y^−1^), and secondary production (mg m^−2^ y^−1^). We obtained the average values for each variable and their standard error from the tables or text of the paper. When the values were presented in figures, we estimated the numerical value using the GetData Graph Digitizer 2.26 software. In some papers, the authors reported mean values without estimates of the variance and these values were retained and used to compute Table 1, although they could not be used for the statistical comparisons, which required an estimate of the variance. A few papers provided values for two or more years (or months) for the same locality or values for two or more nearby habitats, without providing an estimate of the variance. In these cases, we calculated the mean values across the times/sites and the standard error of the mean and used these values for our meta-analysis. In the very few cases where the estimates of the SE were zero (because they were based on several identical values) or extremely low, we bounded the SEs to 0.2855 for density, 0.4560 for biomass, and 1.2780 for secondary production to avoid giving all the weight to just a few studies with a very low SE. These SE values were chosen to be the 15th percentile of the SE distribution.

To analyze the patterns of density, biomass, and secondary production in relation to the family, suborder, habitat, latitude, longitude, and altitude, we used Linear Mixed Models (LMMs) with weights inversely proportional to the variances of the estimates to account for the different precision expected for each study. Given the possibility that temperate streams may have a higher secondary productivity than tropical ones [30], we included a quadratic term for latitude. As a random factor we included the experiment to account for the variability related to each study, because in some cases several data were obtained for the same or different species from a single paper. The response variables were transformed (log(x + 1)) to meet normality. We started with the most complex model, including all the predictor variables. In a second step, we removed all the variables with a *p*-value higher than 0.2 and estimated the effects from this reduced model. If further variables showed a *p*-value higher than 0.2, the process was repeated. The Akaike information criterion was calculated, and the model with a lower AIC was retained [31]. All the statistical analyses were performed with Genstat 24th edition (www.vsni.co.uk), xlStat (www.xlstat.com), and Microsoft Excel. The values are presented as the mean ± SE (and the sample size, if not stated in Table 1). 

## 3. Results

Overall, we found 29 datasets for the Odonata as a whole and 199 datasets concerning the density, biomass, or secondary production for 109 species of odonates belonging to seventeen families (nine Zygoptera and eight Anisoptera) (Table 1). In a further 44 cases, the dataset was identified to the genus, family, or suborder level.

For the studies that identified the species, the geographical distribution was biased to North America (in twenty-four studies), Central/South America (in five studies), Europe (in three studies), and Asia (in four studies). Australia (one study) and Africa (one study) were the least studied regions.

### 3.1. Density

Odonata larvae are abundant in freshwater systems, with a mean density of 240.04 ± 48.01 individuals m^−2^ for the whole order (Table 1). At the suborder level, the larval density is higher in Zygoptera (124.40 ± 76.29) than in Anisoptera (35.39 ± 8.16; Table 1). At the family level (Table 1), Coenagrionidae shows the highest density with 170.90 ± 115.50 individuals m^2^ and Macromiidae the lowest with only 0.25 individuals m^−2^ (only one dataset was available for this family).

We analyzed the effect of the suborder, the family inside the suborder, habitat, latitude, latitude^2^, altitude, and longitude on the density of larval odonates, including the study as a random term, with the weights inversely proportional to the variance of the estimates, using a LMM. This model had an AIC = 638.88. The latitude, latitude^2^, altitude, and longitude had *p*-values higher than 0.2 and were therefore removed. The reduced model had an AIC = 592.58 and was consequently selected. The results indicate that the suborder, family, and habitat affect the density of larval odonates (Table 2). The average density of Zygoptera is higher than that of Anisoptera (Table 1). Lentic habitats show much higher densities (104.40 ± 55.31 individuals m^−2^, N = 118) than lotic systems (27.12 ± 5.09, N = 70).

### 3.2. Biomass

The biomass estimations for odonates indicate a value of 488.56 ± 134.51 mg m^−2^ y^−1^ (Table 1). The mean value for Zygoptera (66.90 ± 26.34) is half that of Anisoptera (141.50 ± 52.44; Table 1), but this difference is not significant (Table 3). The families with the highest biomass were Aeshnidae (202.70 ± 102.30), Corduliidae (192.80 ± 117.00), and Gomphidae (182.90 ± 166.90), with all of them belonging to the Anisoptera suborder (Table 1).

Using a similar LMM as above, we found that the full model had an AIC = 187.59. We dropped the family, latitude, latitude^2^, habitat, altitude, and longitude. The reduced model, with an AIC of 166.96, does not contain any factor explaining a significant amount of variability (Table 3). The mean biomass in lentic habitats was 87.32 ± 24.68 mg m ^−2^ y^−2^ (N = 34), similar to that found in lotic systems, which averaged 128.70 ± 59.15 (N = 31).

### 3.3. Secondary Production

The annual secondary production for odonates averages 3558.02 ± 2146.80 mg m^−2^ y^−1^, with a mean of 536.80 ± 216.80 for Zygoptera and 657.50 ± 216.80 for Anisoptera. The analysis using the full LMM has an AIC of 180.47. The habitat, latitude, latitude^2^, altitude, and longitude had *p*-values higher than 0.2, which were removed. The reduced model had an AIC = 144.04 and was therefore selected. This model indicates that the suborder and family have a significant effect on the secondary production of odonates (Table 4). The annual secondary production of lentic systems is similar (747.50 ± 330.60, N = 25) to that in lotic systems (506.10 ± 210.10, N = 34).

## 4. Discussion

Our review indicates that larval odonates show a mean density of about 240 individuals m^−2^ (Table 1). Wesner [23] summarized the relationship between the stream temperature and benthic macroinvertebrates for the American continent and found that predatory insects (mainly odonates) had a density of 69.4 ± 14.97 ind m^−2^ (±SE; a mean of 78 in temperate sites and 61.5 in tropical sites), which is lower than the values we found. As expected, given the difference in size, zygopterans are more abundant than anisopterans (Table 1). The ratio between both suborders is used as an indicator of ecosystem integrity in tropical streams [32] and is therefore of practical importance. The different families show contrasting densities, with a maximum of ~6500 individuals m^−2^ for the coenagrionid *Telebasis salva* [33], a value that is exceptional due to the thermally constant nature of the stream where the population was studied. However, a further six species show densities over 300 individuals m^−2^. Among the anisopterans, the maximum densities are reported for the Libellulidae/Corduliidae (243 ind m^−2^ [34]), Aeshnidae (200 ind m^−2^ for *Rhionaeshna marchali* [35]), and Libellulidae (338–382 ind m^−2^ [36]) families.

The biomass estimates for larval odonates average 488.56 mg m^−2^ y^−1^ (Table 1), which is higher than the values reported for predatory macroinvertebrates in American streams (99.5 ± 24.27 mg m^−2^ y^−1^ [23]). In this case, the biomass is higher for anisopterans (twice as high), even if their density is lower, due to the larger size of dragonflies compared to damselflies, but the difference is not significant, due to the large variability of the dataset (Table 3). The mean biomass of anisopterans (141.50 ± 52.44) is in the higher range for stream studies. For instance, in a Costa Rican stream, the biomass of aquatic insects ranged from 10 to 36 mg m^−2^ y^−1^, but, in this case, one coenagrionid (*Argia*) contributed up to 47% of the biomass in the riffle microhabitats [37]. The biomass of odonates does not show a relation to latitude (Table 3).

The annual production of macroinvertebrate communities in streams worldwide ranges from 1 to 1000 g m^−2^ [38]. Diptera, Trichoptera, and Ephemeroptera are the orders that show the maximum abundance in streams. Our review indicates that the secondary productivity of odonates averages ~3558 mg m^−2^ y^−1^, a value that is in the lower range of the above interval. However, the maximum reported productivity for odonates is 36.85 g m^−2^ in a mesotrophic lake [39]. The mean value is highly biased upwards by this and other several extraordinary values, and further research is, therefore, needed. The low values for odonates are not surprising, considering that they are top predators and, therefore, the amount of energy available for this trophic level limits their productivity. The total secondary productivity is slightly higher for dragonflies than damselflies, but the maxima are higher for damselflies (Table 1).

Jacobsen et al. [40] reviewed the density, biomass, and secondary production of aquatic insects from tropical streams but did not include odonates. However, they contribute in a significant way to ecosystem processes due to their large size, because secondary production is highly correlated and dependent on biomass, and both density and body size are the principal determinants of biomass [41]. As Jacobsen et al. [40] indicate, the secondary production of aquatic insects in tropical streams is not especially high, due to their low density and body size [42,43]. Groups of macroinvertebrates that are in the lower levels of the trophic chain, like trichopteran filter feeders, range from 0.19 mg m^−2^ y^−1^ to 2.27 g m^−2^ y^−1^ (Table VII in [40]), values that are in the lower range for filter-feeding macroinvertebrates in subtropical streams (3–300 g m^−2^ y^−1^ [38]) and lower than the values reported for odonates (Table 1). Our review does not support an effect of latitude on the abundance or biomass of odonates (Table 1, Table 2 and Table 3), but some of the maximum values are found at around 30–40º of latitude. This factor should be reanalyzed when more studies become available. Of particular relevance would be more studies in Africa and Asia, regions that are understudied.

In conclusion, our review indicates that the amount of matter and energy potentially exported by odonates to terrestrial ecosystems is significant and can therefore contribute to an increase in the productivity of terrestrial ecosystems [21], but they can also affect some ecosystem services via their predation on pollinators [44]. On the other hand, a recent study has shown that odonates can also be pollinators of aquatic plants [45] and in this way contribute to another ecosystem service. Details of the amounts of matter potentially exported can be found in the literature compiled in the Appendix A, where the original papers are listed [46,47,48,49,50,51,52,53,54,55,56,57,58,59,60,61,62,63,64,65,66,67,68,69,70,71,72,73,74,75,76,77,78,79,80,81,82,83,84]. Adult odonates have powerful flight and can move and disperse over large distances from their emergence sites [28], in contrast to other aquatic insects like Ephemeroptera, Plecoptera, and Trichoptera, which remain close to water courses. Finally, odonates are very important as food items for vertebrates, including humans in many countries around the world and are among the most conspicuous insects, a fact that explains their wide use as model insects in research but also in human culture.

## Figures and Tables

**Table 1 insects-15-00510-t001:** The mean values and estimates of the variability (SE, standard error) of the density, biomass, and secondary production of odonates by suborder and family. The values for Odonata came from 29 studies that analyzed the whole order. The other values are from 38 papers that identified the taxa to lower taxonomic levels.

	Density (Individuals m^−2^)					Biomass (mg m^−2^ y^−1^)					Secondary Production (mg m^−2^ y^−1^)				
Taxon	Mean	SE	N	Min	Max	Mean	SE	N	Min	Max	Mean	SE	N	Min	Max
Odonata	240.04	48.01	25	0.50	2539.48	488.56	134.51	11	6.78	1256.50	3558.02	2146.80	17	4.32	36,850.00
Zygoptera	124.40	76.29	85	0.21	6486.00	66.90	26.34	30	0.45	783.40	536.80	216.80	24	0.84	7905.00
Anisoptera	35.39	8.16	103	0.12	558.00	141.50	52.44	35	0.09	1682.00	657.50	216.80	35	1.33	6842.00
Calopterygidae	31.32	12.17	7	0.33	92.57	36.23	35.77	2	0.47	72.00	237.10	236.20	2	0.84	473.30
Chlorocyphidae	1.20		1												
Coenagrionidae	170.90	115.50	56	0.21	6486.00	86.18	38.85	20	1.00	783.40	642.60	461.40	17	1.19	7905.00
Euphaeidae	28.31	4.91	2	23.40	33.22	33.75		1			694.90	545.50	2	149.40	1240.00
Lestidae	27.65	17.29	10	0.70	172.10	34.72	12.96	5	2.60	73.00	76.80		1		
Philogangidae	0.40		1												
Platycnemididae	111.30	62.84	4	4.70	277.70										
Platystictidae	0.91	0.15	2	0.76	1.05	0.45		1			2.66		1		
Pseudostigmatidae	0.65		1												
Aeshnidae	30.71	13.38	20	0.22	200.00	202.70	102.30	7	30.50	791.30	290.60	121.90	8	6.78	794.60
Cordulegastridae	9.15	6.15	2	3.00	15.30	90.33		1			682.70	367.30	2	315.30	1050.00
Corduliidae	6.64	3.03	8	0.70	24.50	192.80	117.00	3	0.42	404.50	880.40	459.60	5	4.73	2022.00
Gomphidae	8.88	2.69	14	0.40	33.33	182.90	166.90	10	0.09	1682.00	902.90	681.40	10	1.33	6842.00
Libellulidae	35.40	10.44	51	0.12	382.80	65.80	24.58	11	0.26	244.00	654.60	375.10	7	1.56	2220.00
Macromiidae	0.25		1			13.80		1			69.00		1		
*Oxygastra* (incertae sedis)	3.80		1												
Synthemistidae	6.70		1			2.20		1			52.20		1		

**Table 2 insects-15-00510-t002:** The results of a Linear Mixed Model with density as the response variable (Log(x + 1) transformed). The final model includes the suborder, the family inside the suborder, and the habitat as fixed effects and the experiment as a random factor. The tests were for fixed effects, sequentially adding terms to the fixed model. The significant variables are in **bold**.

Fixed Term	Wald Statistic	n.d.f.	F Statistic	d.d.f.	F pr
**Suborder**	15.76	1	15.76	131.1	**<0.001**
**Suborder.Family**	56.19	15	3.69	112.1	**<0.001**
**Habitat**	5.41	1	5.41	30.4	**0.027**

**Table 3 insects-15-00510-t003:** The results of a Linear Mixed Model with biomass as the response variable (Log(x + 1) transformed). The final model includes suborder as a fixed effect and the experiment as a random factor. The tests were for fixed effects, sequentially adding terms to the fixed model. The model is not significant.

Fixed Term	Wald Statistic	n.d.f.	F Statistic	d.d.f.	F pr
Suborder	1.68	1	1.68	49.0	0.201

**Table 4 insects-15-00510-t004:** The results of a Linear Mixed Model with secondary production as the response variable (Log(x + 1) transformed). The final model includes the suborder and family as fixed effects and the experiment as a random factor. The tests were for fixed effects, sequentially adding terms to the fixed model. The significant variables are in **bold**.

Fixed Term	Wald Statistic	n.d.f.	F Statistic	d.d.f.	F pr
**Suborder**	20.65	1	20.65	27.6	**<0.001**
**Suborder.Family**	102.26	9	10.91	27.6	**<0.001**

## Data Availability

No new data were created or analyzed in this study. Data sharing is not applicable to this article.

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
