# Peer review of "A Review of the Density, Biomass, and Secondary Production of Odonates"

_insects, 2024, doi:10.3390/insects15070510_

Round 1

Reviewer 1 Report

Comments and Suggestions for Authors

Congratulations to the authors on this manuscript. It is well-written, synthesised, and easy to follow. The topic is interesting, and the reviews help summarise the sparse knowledge across different publications. I recommend the publication of the paper with only a few minor revisions, detailed below.

Specific comments (by lines)

Line 8: Replace the semicolon with a colon.

Line 16: Could you check the structure of this sentence? The construction is a bit strange, and I’m not quite sure I grasp the meaning.

Line 90: You mention 34 papers selected from... how many did you find in your initial search?

Supplementary material: Please change “Table S1” to “Supplementary Material 1” or something similar. The use of "table" initially led me to expect a table, and it wasn't until I read the description at the end of the document that I understood it referred to the entire Excel document, rather than a mislabelled Excel tab, as I had initially thought.

Line 95: The longitude is in absolute value as well, right? Please rewrite the sentence as it currently confuses the reader.

Line 97: Google Earth, like other software, needs to be cited properly. You can find the citation format here: https://libguides.northampton.ac.uk/harvard/examplesJ-M/mapgoogleearth

Lines 110–111: Perhaps I lack sufficient expertise, but why use the 15th percentile? Could you provide a reference?

Lines 112–113: The authors have used uppercase letters in the list of response and explanatory variables. Please consider using correct English, without uppercase letters.

Line 120: Why 0.1?

Lines 124–125: Please check the correct citation format for the software used.

Line 125: If “Table” does not refer to a specific table, please write the word in lowercase.

Lines 140–141: The authors have used uppercase letters in the list of variables. Please consider using correct English, without uppercase letters.

Line 140: Why did you analyse the effect of family within the suborder? I found no mention of this in the Methods section. If you also analysed the interaction between variables, could you include a sentence explaining this in the Methods section?

Lines 140–145: You have explained in the Methods the steps for selecting the variables and the final model, so you can simplify this paragraph by avoiding explanation of the AIC and the variables of the non-selected model.

Table 2: As mentioned in line 146 where you discuss significance, could you indicate or mark the significant values in Table 2?

Figure 1: Could you label each graph with letters, e.g., "a) density, b) biomass…” and explain them in the figure footnote? Also, in the Y-axis, the 2 of squared meters needs to be in superscript.

Lines 168–171: As before, since you have explained the process of model selection, I would suggest avoiding repetition. For example, you could reduce the mentioned lines to “The difference in AIC between the models is negligible…”.

Table 3: As mentioned in line 174 regarding significance, could you indicate or mark the significant values in Table 3?

Figure 2: On the Y-axis, the 2 of squared meters needs to be in superscript.

Lines 185–187: As before, since you have already included the model selection process, these lines can be removed; directly explain the selected model instead.

Table 3: As in line 188, where you talk about significance, could you indicate or mark the significant values in Table 3?

Lines 197–198: This sentence is identical to that in line 16. Could you check the structure?

Lines 197–211: Please consider moving these two paragraphs to the Introduction. They do not include any data or analysis of your results and would be more contextually appropriate in that section.

Line 248: Correct “Insects” to lowercase as it does not need to begin with an uppercase letter here.

Line 253: Include a full stop after the reference, as you are starting a new sentence.

Line 257: Perhaps after "odonates," include a reference to Table 1?

Reviewer 2 Report

Comments and Suggestions for Authors

Evaluation of the manuscript “A review of density, biomass and secondary production of odonates“

     This is a hot topic, interesting for a wide range of readers. Therefore, I definitely support to publish it in the journal. The information collected and presented in the paper is long overdue, and indeed it is a useful contribution to the ecosystem functioning, focusing on biomass production of odonates.

     Dragonflies and damselflies are especially important group of organisms in the functioning of ecosystem, because of their amphibian character (i.e., they are both related to the water and the terrestrial part of a landscape). This is an important aspect of this taxa highlighted in the paper of Nagy et al. (doi.org/10.1038/s41598-019-54628-7). I propose to cite it. Moreover, odonates are really important bioindicator taxa; it is a notable characteristic of odonates that they are good ecological indicators for both aquatic and terrestrial habitat quality. However, it is an especially notable, special feature of this taxa that the exuviae can be used as an environmentally friendly method for environmental impact assessment without killing specimens (see. Simon et al. DOI: 10.1111/eea.13412); thus, it can be used in case of protected insect populations (doi.org/10.3390/w11112200).

     The manuscript is generally well written, easy to understand, although the English usage should improve. The Simple Summary is an advertisement of the taxa. I propose to write an informative Simple Summary based on the real value (message) of the manuscript.

     The Abstract is more informative than the Simple Summary, but the first half of it is also just a hype-of-the-topic. I propose to revise both the Abstract and the Simple Summary.

 Tables are useful and informative.

     Figures are terrible. You should think to revise them completely. There is no reason to write the equation of the linear relationship to the drawing area. R2 also should be in the Figure legend; although these values are very low. This makes the (linear) relationship rather questionable. In case of Figure 1 I propose to think about eliminating the outliers or to use robust line-fitting according to Tukey-style (included into R packages).

     In case of Figure 2 evidently there is an extreme outlier; thus, you should omit it. The relationship is evidently nonlinear. You should try to use logarithmic x-axis.

Comments on the Quality of English Language

Evaluation of the manuscript “A review of density, biomass and secondary production of odonates“

This is a hot topic, interesting for a wide range of readers. Therefore, I definitely support to publish it in the journal. The information collected and presented in the paper is long overdue, and indeed it is a useful contribution to the ecosystem functioning, focusing on biomass production of odonates.

Dragonflies and damselflies are especially important group of organisms in the functioning of ecosystem, because of their amphibian character (i.e., they are both related to the water and the terrestrial part of a landscape). This is an important aspect of this taxa highlighted in the paper of Nagy et al. (doi.org/10.1038/s41598-019-54628-7). I propose to cite it. Moreover, odonates are really important bioindicator taxa; it is a notable characteristic of odonates that they are good ecological indicators for both aquatic and terrestrial habitat quality. However, it is an especially notable, special feature of this taxa that the exuviae can be used as an environmentally friendly method for environmental impact assessment without killing specimens (see. Simon et al. DOI: 10.1111/eea.13412); thus, it can be used in case of protected insect populations (doi.org/10.3390/w11112200).

The manuscript is generally well written, easy to understand, although the English usage should improve. The Simple Summary is an advertisement of the taxa. I propose to write an informative Simple Summary based on the real value (message) of the manuscript.

The Abstract is more informative than the Simple Summary, but the first half of it is also just a hype-of-the-topic. I propose to revise both the Abstract and the Simple Summary.

Tables are useful and informative.

Figures are terrible. You should think to revise them completely. There is no reason to write the equation of the linear relationship to the drawing area. R2 also should be in the Figure legend; although these values are very low. This makes the (linear) relationship rather questionable. In case of Figure 1 I propose to think about eliminating the outliers or to use robust line-fitting according to Tukey-style (included into R packages).

In case of Figure 2 evidently there is an extreme outlier; thus, you should omit it. The relationship is evidently nonlinear. You should try to use logarithmic x-axis.

Reviewer 3 Report

Comments and Suggestions for Authors

Rivas-Torres and Cordero-Rivera reviewed the literature on larval Odonata biomass, density, and secondary production. Research on the factors influencing these topics at a large-scale, like was attempted in this paper, would provide an important contribution to the field and serve as a baseline for comparison with future work. However, I found this paper to have major flaws the framing and methodology. Additionally, there were numerous inconsistencies that need to be resolved (e.g., number of studies in abstract differs from main text; value reported on l. 150 [6,500 ind m-2] does not appear in figures).

Framing: The review is placed in the context evaluating the contribution of odonates to ecosystem linkages between adjacent aquatic and terrestrial ecosystems (e.g., l. 13-14; l. 26-29; l. 258-261). Research on the transport of nutrients, biomass, and resources across ecosystem boundaries by emergent aquatic insects is an important field for which much work is still needed. I do not doubt that is possible that odonates may contribute substantially to the export of materials from aquatic ecosystems and to dynamics on the surrounding terrestrial landscape. However, I do not think the data presented provides sufficient evidence to make the claims the author presented.

As far as I can tell, the study only concerns the dynamics of larval odonates. While this is important context for the dynamics of aquatic ecosystems, it is known that not all aquatic production enters terrestrial systems For example, a review by Gratton and Vander Zanden 2009 (https://doi.org/10.1890/08-1546.1) found that ratio of emergence to production varies widely, but is most often below 0.3 (Fig 2C). To more accurately estimate the potential contribution of aquatic insects to terrestrial landscapes, data on emerging individuals would better characterize odonate contributions to the export of material.

Evaluating the contribution of one taxa of emergent aquatic insect to the export of material from streams requires a comparison. In the absence of data on other emergent aquatic insects, stream nutrient conditions, and potentially the background productivity of comparable resources in the adjacent ecosystem (Marczak et al. 2007 https://doi.org/10.1890/0012-9658(2007)88[140:MTLHAP]2.0.CO;2), the absolute values presented in this study are not meaningful to me.  

Methodology: The methodology employed in this study is presented in insufficient detail to properly evaluate the claims in the manuscript, the study lacks important context, and some of the statistical methods employed risk type I errors.

There are important details of a review that are missing. Just a few examples:

How many papers were reviewed? What were the selection criteria? It seems unlikely that 34 papers represent all papers that characterize biomass, abundance, or secondary production in Odonata.

What methods were used in the papers to calculate biomass, production, and density? For example, are lots of different formulations to calculate secondary production (Benke and Huryn 2017 https://doi.org/10.1016/B978-0-12-813047-6.00013-9). I think it’s important to, at a minimum, discuss which metrics odonatologists use and to discuss how comparable or not they are. Similarly, it is important to discuss sampling methodology.  Are Ekman grabs comparable with Hess samplers (or other quantitative sampling designs)? Are all density/biomass estimates taken during peak biomass?

What was the spatial distribution of studies included in the paper? It would be useful to use these data to identify data poor regions and gaps in the literature. A map of study locations and a description of the types of regions sampled is important. (e.g., do samples mostly come from riffles in small order streams and large lakes? Are there microhabitats that are underrepresented?)

Which taxa are most represented? Are the studies in this paper characteristic of global odonate diversity or biased by taxa studied by a few influential labs? Additionally, this opens up questions about why some taxa are more abundant/productive than others. This is a great opportunity to explore more in the manuscript.

The model selection approach (including only factors that have a p-value <0.1) has serious problems with increasing type I error. If this approach is taken, p values should not be presented in the text. Based on the figures, I am very suspicious about such low p values for latitude and density (for example).

I think the figures ought to be on the same scale as the analyses, include their uncertainty estimates, and should not have regression lines unless the relationship is significant in the model. The R2 values presented suggest that the relationships presented, even if significant, are biologically meaningless with none of the variables explaining more than 1% of the variance. I would prefer the authors report this negative result, as it is much more interesting.

I was also unsure why we would expect a linear relationship between latitude and the observed variables. It seems reasonably likely that temperate latitudes have the most productive macroinvertebrates, as has been reported in marine benthic macroinvertebrates (e.g., Cusson et al. 2005; doi:10.3354/meps297001).

Comments on the Quality of English Language

The manuscript is legible and coherent. I had no major concerns about the quality of the English used by the authors. 
